**www.cambridge.org/qrd**

# On the possibility of carbon-free heteropolymers on Venus: a computational astrobiology study

Ishaan Madan[1,2] ⓘ, Shekoufeh Arabi Aliabadi[1] ⓘ, Johanna Huhtasaari[1],
Ebba Matic[1], Emil Hogedal[1], Kinga Kamińska[1], Filip Nilsson[1], Axel Stark[1],
Fernando Izquierdo-Ruiz[1], Hilda Sandström[1] and Martin Rahm[1] ⓘ

[1]Department of Chemistry and Chemical Engineering, Chalmers University of Technology, Gothenburg, Sweden and
[2]Department of Earth, Atmospheric, and Planetary Sciences, Purdue University, West Lafayette, IN, USA

## Research Article

**Keywords:**
abiotic polymerization; condensed-phase astrochemistry; prebiotic chemistry; sulfuric acid chemistry; Venus clouds

**Corresponding author:**
Martin Rahm;
Email: martin.rahm@chalmers.se

Fernando Izquierdo-Ruiz, Current address: Department of Physical Chemistry, Complutense University of Madrid, Madrid, Spain.

Hilda Sandström, Current address: Department of Applied Physics, Aalto University, Espoo, Finland.

## Abstract

This work poses and partially explores an astrobiological hypothesis: might polymeric sulfur and phosphorus-based oxides form heteropolymers in the acidic cloud decks of Venus' atmosphere? Following an introduction to the emerging field of computational astrobiology, we demonstrate the use of quantum chemical methods to evaluate basic properties of a hypothetical carbon-free heteropolymer that might be sourced from feedstock in the Venusian atmosphere. Our modeling indicates that R-substituted polyphosphoric sulfonic ester polymers may form via multiple thermodynamically favorable pathways and exhibit sufficient kinetic stability to persist in the Venusian clouds. Their thermodynamic stability compares favorably to polypeptides, whose formation is slightly thermodynamically unfavored relative to amino acids in most known abiotic conditions. We propose a combined approach of vibrational spectroscopy and mass spectrometry to search for related materials in Venus's atmosphere but note that none of the currently planned missions are well suited for their detection. While predicted Ultraviolet–Visible spectra suggest that the studied polymers are unlikely candidates for Venus's unidentified UV absorbers, the broader possibility of sulfuric acid–based chemistry supporting alternative biochemistries challenges the traditional carbon-centric models of life. We argue that such unconventional lines of inquiry are warranted in the search for life beyond Earth.

## Introduction

Earth's biochemistry is carbon-based, dependent on liquid water, and fueled by various energy sources, including sunlight and geothermal gradients. Beyond Earth, different environmental conditions could favor entirely distinct frameworks for life, raising questions such as: if not carbon-based, and not dependent on liquid water, what alternative chemistries and solvents might facilitate the emergence of complex, life-like systems? Tackling such a question scientifically is not an easy exercise, for it is inherently speculative. In this work, we entertain one such possibility, arguing for the potential of sulfur and phosphorus to supplement the role of carbon in forming alternative heteropolymers in the acidic cloud decks of Venus' atmosphere. To investigate this hypothesis, we present a computational astrobiology case study of an R-substituted polyphosphoric sulfonic ester (Figure 1A).

Venus has long intrigued scientists as a potential site for extraterrestrial life, despite its extreme surface conditions (Bains et al., 2021, 2023; Cockell, 1999; Dartnell et al., 2015; Grinspoon and Bullock, 2007; Izenberg et al., 2021; Kotsyurbenko et al., 2021; Limaye et al., 2018, 2021; Mogul et al., 2021; Morowitz and Sagan, 1967; Patel et al., 2022; Petkowski et al., 2024; Schulze-Makuch et al., 2004; Seager et al., 2021), paving the way for numerous future missions that are in development or on the horizon (Kotsyurbenko et al., 2024): NASA's DAVINCI mission (tentative: 2030) (Garvin et al., 2022), MIT/Rocket Lab's "Morning Star" (no earlier than 2026 (French et al., 2022; Seager et al., 2022), ESA's EnVision (2031) (de Oliveira et al., 2018), NASA VERITAS Orbiter (no earlier than 2031) (Smrekar et al., 2022), ISRO's Shukrayaan-1 (potentially 2026) (Sundararajan, 2021), and Roscosmos Venera-D (proposed 2030s) (Zasova et al., 2019).

Venus's dense atmosphere is primarily composed of carbon dioxide (~96.5%) and nitrogen (~3%) with traces of other gases, including sulfur dioxide and water vapor (Hoffman et al., 1980; Johnson and de Oliveira, 2019), leading to surface pressures nearly 90 times that of Earth and temperatures exceeding 460 °C (Taylor et al., 2018). However, at 48–70 km altitude, Venus's atmospheric cloud decks offer Earth-like pressures and temperatures, creating a potential environment for liquid solvents (S. S. Limaye et al., 2018). Venusian clouds, rich in sulfuric acid aerosols, host complex chemistry, and unknown UV absorbers, making them a subject of astrobiological interest (Esposito et al., 2022a; Titov et al., 2018). Recent, though debated,

**Figure 1.** A) An R-substituted poly (phosphoric sulfonic ester), a hypothetical class of Venusian heteropolymers imagined mimicking the structure of polypeptides. B) An R-substituted polypeptide, representative of the building blocks of life as we know it.

potential detections of phosphine (Greaves et al., 2021, 2020) and ammonia (Bains et al., 2023, 2024; Duzdevich et al., 2024; S. Seager et al., 2024) further highlight Venus as a testbed for Earth-like and unconventional biochemistries (Bains et al., 2023, 2024; Duzdevich et al., 2024; S. Seager et al., 2024).

The primary component of Venus' dense cloud layers is inferred to be sulfuric acid (Bains et al., 2023). This liquid is a reactive, polar protic solvent that exhibits strong hydrogen bonding networks and readily exchanges protons with dissolved molecules. These features, alongside its natural occurrence, solvation capacity, chemical stability, and chemical functionality, are criteria that can be used to assess how life-sustaining an alternative solvent might be (see (Bains et al., 2024) for a detailed discussion). While sulfuric acid meets most such criteria and compares favorably with water, it can readily degrade many organic compounds (e.g., sugars and dipeptides (Petkowski, Seager, Bains, et al., 2024)). However, recent preliminary work has demonstrated the extended persistence of amino acids (M.D. Seager et al., 2024) and nucleic acids (S. Seager et al., 2024) in concentrated sulfuric acid. Because the ability of sulfuric acid to decompose terrestrial biomolecules has not been systematically investigated (Bains et al., 2023), it remains largely unknown which forms of biochemistry this liquid might support. It is worth noting that water is also highly reactive under many conditions, e.g., in hydrolysis or in acid–base catalysis, depending on pH, and any life utilizing sulfuric acid as a solvent would presumably have evolved to thrive in it. We approach this possibility by examining the potential formation and stability of polymeric sulfur- and phosphorus-based oxides, which could maintain structural and functional stability in acidic media.

Beyond its scientific motivation, this project also served as a platform to engage and train successive undergraduate researchers. Their substantive contributions across multiple stages of the work are reflected in the large author list.

## The chemical flexibility of sulfur and phosphorus

### *Sulfur and phosphorus in terrestrial biochemistry*

Life on Earth is shaped by the abundance of reactive elements in the universe, particularly carbon, hydrogen, nitrogen, oxygen, phosphorus, and sulfur (CHNOPS), in some combination with other elements (Domagal-Goldman et al., 2016). Carbon is central to the CHNOPS elements, forming the covalent backbone of life's macromolecules: carbohydrates, lipids, proteins, and nucleic acids, all optimized for liquid water. The polypeptide chain illustrated in Figure 1B is one such carbon-rich example that relates to the imagined function of the R-substituted poly (phosphoric sulfonic ester) we study in this work.

Figure 2A shows 3′-phosphoadenosine-5′-phosphosulfate (PAPS), another example wholly representative of CHNOPS. PAPS is the most common coenzyme facilitating sulfotransferase reaction (Günal et al., 2019) and is highlighted here as a demonstration of the existence of –S–O–P–linkages in biology. Polyphosphates and phosphate esters, a motif related to our target of inquiry, are ubiquitous in biology, being involved in, e.g., signal transduction, energy production and transfer (e.g., Adenosine triphosphate/diphosphate (ATP/ADP)), and the regulation of protein function (Westheimer, 1981). Both sulfur and phosphorus have numerous other roles in biology, ranging from structural components such as in sulfo- and phospholipid membranes, to metabolic intermediates, to signaling and enzymatic cofactors (Cleland and Hengge, 2006), and in our genetic code, ribonucleic acid (RNA) and deoxyribonucleic acid (DNA) (Table 1).

### *Sulfur and phosphorus in extreme and Venus-like environments*

Sulfur and phosphorus play critical roles across a variety of life forms, especially in extremophiles that thrive in harsh environments. The capacity of sulfur to hold multiple oxidation states can support energy metabolism and redox buffering, particularly in sulfur-rich or fluctuating conditions (Schulze-Makuch and Irwin, 2006; Wasmund et al., 2017). For example, marine sediment microbes and phototrophic bacteria oxidize sulfur compounds for energy, demonstrating sulfur's adaptability across environmental extremes (Rampelotto, 2010). In acidophiles inhabiting volcanic lakes or acid mine drainage, sulfur compounds play dual roles in energy generation and mitigating acidity through redox buffering (Baker-Austin and Dopson, 2007). Phosphorus, primarily in the form of phosphates and polyphosphates, complements sulfur by providing energy storage, stabilizing cellular structures, and maintaining pH homeostasis. One likely reason for the prevalence of phosphate esters in biology is their substantial kinetic stability. This stability is particularly pronounced when the phosphates are negatively charged, as the resulting electrostatic repulsion hinders nucleophilic addition of anions (Kamerlin et al., 2013). While we do not study kinetic stability in detail in this work, we note that an arbitrary number of negatively charged –OR groups (e.g., alkoxy groups) on a hypothetical R-substituted poly (phosphoric sulfonic ester) could render it negatively charged, thereby affecting the polymer's resistance to hydrolysis and other degradation pathways. Both sulfur and phosphorus are necessary for acid resistance in biology, stabilizing genetic materials like DNA (deoxyribonucleic acid) and RNA (ribonucleic acid) and enabling microbial life in acidic, oxidative environments (Baker-Austin and Dopson, 2007; Barrie Johnson and Hallberg, 2009).

Despite the centrality of phosphate-bearing molecules in modern biology, their prebiotic incorporation on early Earth posed

**Figure 2.** Selected examples of sulfur and phosphorus in biology (A-C), and in Venus-relevant conditions (D-E): A) 3′-phosphoadenosine-5′-phosphosulfate (PAPS), the most common coenzyme in sulfur-group transfer reactions. B) Sulfoquinovosyl diacylglycerol (SQDG), a sulfolipid found in many photosynthetic organisms (Karvansara et al., 2023). C) A general phospholipid, where the R group can be replaced to form major components of cell membranes (e.g., phosphatidylcholine). D) The cyclic trimer phase of sulfur trioxide ($\gamma$-SO$_3$). E) Polymeric phase of sulfur trioxide ($\alpha$-SO$_3$).

**Table 1.** Comparison of carbon, sulfur, and phosphorus

| | Carbon | Sulfur | Phosphorus |
|---|---|---|---|
| **Electronegativity**[a] | 13.9 | 13.6 | 12.8 |
| **Common oxidation States** | −4, −3, −2, −1, 0, +1, +2, +3, +4 | −2, −1, 0, +1, +2, +3, +4, +5, +6 | −3, −2, −1, 0, +1, +2, +3, +4, +5 |
| **Number of bonds** | 1–4 | 1–6 | 1–5 |
| **Multiple allotropes** | Yes | Yes | Yes |
| **Polymeric oxides** | Primarily stable at high pressure, e.g., as poly-CO$_2$ or poly-CO. | Yes, e.g., as $\alpha$-SO$_3$ shown in Figure 2E. | Yes, as metastable polymeric phosphorus pentoxide phases. |
| **Biological significance** | Present, e.g., in backbone of carbohydrates, lipids, proteins, and nucleic acids, and in most molecular biochemistry. | Present, e.g., in proteinogenic amino acids like methionine and cysteine, homocysteine, taurine, vitamins (B7, B1), coenzymes, sulfolipids, protein disulfide linkages, and certain metalloproteins. | Present, e.g., in DNA, RNA, ATP, ADP, several coenzymes and cofactors, and in phospholipids. |

[a]Electronegativity in eV/e$^-$ from (Rahm et al., 2019)

considerable challenges due to their poor solubility in aqueous conditions and the endergonic nature of phosphorylation reactions. This issue, often referred to as the "phosphate problem", is closely linked to the geochemical conditions of the Hadean oceans: high water activity, near-neutral pH, and apatite-dominated reservoirs (see (Pasek et al., 2017) for details). It has been shown (Toner and Catling, 2020) that phosphate can accumulate to high concentrations in carbonate-rich lakes, adding nuance to how the phosphate problem may be alleviated under certain conditions. Phosphate limitations can also be eased under low water activity, strongly acidic or oxidizing conditions, or when transient, highly reactive phosphate phases (e.g., poly- and metaphosphates) are continuously regenerated (Pasek et al., 2017). Venus's sulfuric-acid cloud decks may satisfy all three criteria: (i) extremely low water activity, (ii) persistent acidity, (iii) potential for sulfur and phosphorus cycling of compounds that could replenish reactive species. In such an environment, fully oxidized phosphorus (rather than reduced), could remain chemically accessible and kinetically stable, thereby supporting our focus on high-valent polymeric sulfur- and phosphorus-based oxides.

Sulfur trioxide ($SO_3$) is likely present throughout Venus's atmosphere (Bierson and Zhang, 2020; Petkowski et al., 2024). $SO_3$ is known to form various condensed-phase structures, including monomers, cyclic trimers ($\gamma$-$SO_3$, Fig. 2D), and linear chain polymers ($\alpha$-$SO_3$, Fig. 2E and $\beta$-$SO_3$). These polymorphs contain strong S–O bonds, allowing $SO_3$ to exist in different forms under various temperatures and pressures (Lovejoy et al., 1962). In Venus' cloud decks at 48–70 km altitudes, temperatures range from ~200 to 350 K (Titov et al., 2018). The exact melting points of the $\gamma$-, $\beta$-, and $\alpha$- $SO_3$ phases vary (Lovejoy et al., 1962) but generally lie below 350 K, implying they could exist in dynamic equilibrium under Venus's cloud conditions. We highlight $SO_3$'s inherent ability to form a variety of polymeric oxides for it is a relevant justification for our speculations regarding sulfur- and phosphorus-based biochemistry in Venus's cloud layer.

In Table 1, we compare some properties of carbon, sulfur, and phosphorus, emphasizing their roles in forming (potential) biopolymer analogs. Carbon, while versatile, transforms into carbon dioxide when fully oxidized, which is relatively inert and unable to (spontaneously) form complex structures. In contrast, sulfur and phosphorus retain the ability to form polymeric structures even in high oxidation states. This ability is particularly relevant in environments like Venusian cloud decks, where acidic and oxidative conditions prevail.

## Computational astrobiology

While this study aims to demonstrate the use of quantum chemical calculations to approach specific chemical questions, such as molecular stability and detection characteristics, it does not aim to definitively answer these questions. Instead, our work is intended to serve as a brief introduction to and advocate for computational astrobiology, an emerging and interdisciplinary field in which questions regarding the origin, evolution, and distribution of life in the Universe are approached using computational methods (Chaban et al., 2002). Although the term computational astrobiology is not widespread, many have utilized computational tools to study the origins of life-related questions. Such exercises, which are beyond the scope of this work to review in full, involve the creative integration of methods from computer science and modeling with knowledge from the physical sciences, such as chemistry, biology, geology, paleontology, physics, planetary science, and atmospheric

sciences, to explore questions regarding life beyond Earth. Andrzej Pohorille, who directed the Center for Computational Astrobiology at NASA Ames Research Center (Gronstal, 2024), is one prominent example of an early contributor to this field. Pohorille's work includes, but is not limited to, the computational study of biomolecular systems (e.g., (Pohorille and Wilson, 1993), statistical mechanics of condensed phases (e.g., (Pohorille and Pratt, 1986), as well as some of the earliest molecular dynamics simulations on supercomputers (Pohorille et al., 1990). This breadth of early contributions exemplifies one challenge of discussing computational astrobiology: the field is exceptionally diverse.

For brevity's sake, we limit our following non-exhaustive summary of literature to a selected few cases in which astrobiological hypotheses related to non-Earth-like chemistry are explored using atomistic modeling. We purposefully exclude more conventional astrochemistry, as well as prebiotic chemistry, and origin of life studies focused on processes on Earth. We refer the interested reader to (Grefenstette et al., 2024), Chapter 9 of the Astrobiology Primer 3.0, which examines alternative biochemistries in more detail beyond the computational lens.

Computational structure generation, e.g., using graph-theoretical algorithms, is one approach for exploring the vast chemical spaces that could support exotic life forms (Meringer and Cleaves, 2017). Stochastic simulations of the formation and growth of molecular networks, as well as reflexively autocatalytic set (RAF) analyses, have demonstrated that alternative biochemical systems can sustain autocatalytic feedback loops, resembling metabolic pathways (Grefenstette et al., 2024; Kun et al., 2008; Xavier et al., 2020). Comparison of calculated solvation free energies of molecules in water and cyclohexane has recently served as a preliminary proof-of-concept for exploring the potential of life in alternate media (Giacometti, 2024). Additionally, xeno-nucleic acids, synthetic alternatives to DNA and RNA (e.g., peptide nucleic acids, hexitol nucleic acids) may have predated traditional nucleic acids in Earth's evolutionary history (Brown et al., 2023; Grefenstette et al., 2024), potentially holding a capacity to support primitive information storage. Quantum chemistry has been used to investigate the thermodynamic stability of conformations of model nucleic acid analogs (Alenaizan et al., 2021). Other notable examples are quantum chemical estimates of oxygen-free molecular biochemistry (Lv et al., 2017), the use of ab initio structure prediction and condensed matter calculations to evaluate light harvesting abilities of HCN-based polymers (Lunine et al., 2020; Rahm et al., 2016), and molecular dynamics simulation and phonon spectra calculations to evaluate kinetic and thermodynamic stability of polarity-inverted cell membranes operable in Titan's cold hydrocarbon lakes and seas (Sandström and Rahm, 2020; Stevenson et al., 2015). These latter studies all exemplify the application of quantum mechanical modeling to investigate materials based on H, C, and N, and their potential astrobiological role in cryogenic environments far removed from those of the Earth. In what follows, we instead computationally explore the plausibility of sulfur and phosphorus-rich materials in warmer environments.

## Results and discussion

### Model description

Figure 3 shows the computational model, **1**, representing a hypothetical poly (phosphoric sulfonic ester). Our model of the polymer is chosen to be cyclic to avoid electronic and structural effects

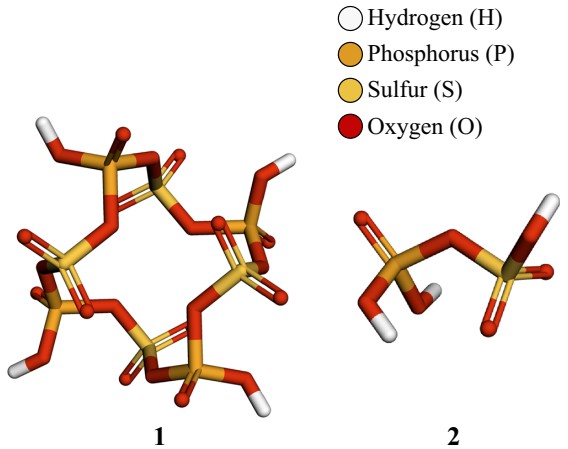

○ Hydrogen (H)
● Phosphorus (P)
● Sulfur (S)
● Oxygen (O)

**1**          **2**

**Figure 3.** Computational models of an R-substituted poly (phosphoric sulfonic ester). 1) S$_4$-symmetric cyclic tetramer. 2) Hypothetical monomer unit, analogous to an amino acid in our comparison with polypeptides.

arising due to end-group functionality that are cumbersome to describe (Sandström et al., 2024). In other words, while the model is molecular it is also intended to emulate a long-chain polymer. The $S4$-symmetric **1**, used for our thermodynamic and vibrational analyses, is devoid of ring strain and exhibits the lowest Gibbs free energy per repeat unit among several screened cyclic oligomers while not featuring intramolecular hydrogen bonding (see Figure S1). It is therefore a reasonable proxy for a linear chain. The hydrogen atoms serve as placeholders for potentially large R-groups.

Also shown in Figure 3 is monomer unit **2**, a hypothetical precursor to the envisioned polymer chain. This structure facilitates estimates of bond strengths and enables a direct comparison with amino acids, the monomer units of polypeptides. However, as we will discuss, **2** need not be the only viable feedstock molecule for forming **1** or related structures.

All structures, energies, and properties are calculated using density functional theory (DFT), combined with algorithms for conformational sampling and methods for implicit consideration of sulfuric acid solvation effects, the details of which are provided in the methods section.

## The thermodynamics of formation of a Venusian polypeptide analog

Polypeptide synthesis is fundamental to biochemistry and proceeds through dehydration, i.e., via the elimination of one water molecule (from the reactants) per reaction step. Le Chatelier's principle tells us that because water is formed, dehydration processes are penalized in aqueous conditions. Indeed, unaided aqueous polymerization of amino acids is thermodynamically unfavored by a positive Gibbs free energy change of 3–5 kcal/mol (Kitadai et al., 2011; Lehninger, 1971), and the question of how prebiotic polypeptide synthesis occurs is still unsolved. In living systems, polypeptides are made through a combination of enzymatic catalysis and the utilization of energy-rich substrates (e.g., ATP). How does our hypothetical phosphoric sulfonic ester polypeptide analog compare?

To find out, we calculated the energetics of sequential insertion of our envisioned monomer **2**, into expanding cyclic polymers akin to **1**, i.e.,

$$2 + (SO_3POOR)_n \rightarrow (SO_3POOR)_{n+1} + H_2O \qquad (1)$$

where $(SO_3POOR)_n$ denotes a cyclic polymer composed of $n$ monomer units. The computed Gibbs free energy of reaction, $\Delta G_r$, is provided for $n$ = 1–4 in Supplementary Table S2. Small cycles such as dimers and trimers are penalized by ring-strain, whereas $\Delta G_r$ approaches $\approx -1$ kcal/mol for larger structures, such as **1**, used here as a proxy for longer chains. Under Venus's cloud-deck conditions, polymer elongation is therefore predicted to be near-thermoneutral to mildly exergonic. This contrasts with peptide oligomerization in aqueous media, which is thermodynamically uphill.

Although the formal condensation chemistry and thermodynamic profiles are broadly analogous, there are notable structural differences: a phosphoric–sulfonic repeat unit has six rotatable bonds in its backbone and first substituents, versus three in a peptide (Figure 1). The increased torsional flexibility, together with the uncertain nature of pendant side chains in the heteropolymer, may be more likely to disrupt backbone regularity.

Our dehydration polymerization in reaction 1 assumes the availability of **2**. But might this hypothetical monomer feasibly form in the Venusian atmosphere? As a limited test of this hypothesis, we evaluate a selection of reactions involving plausible phosphorous- and sulfur-bearing feedstock, $H_3PO_4$, $SO_3$, or $HPO_3$:

$$H_3PO_4 + H_2SO_4 \rightarrow \mathbf{2} + H_2O \qquad \Delta G_r^0 = 1 kcal/mol \qquad (2)$$

$$H_3PO_4 + SO_3 \rightarrow \mathbf{2} \qquad \Delta G_r^0 = -3 kcal/mol \qquad (3)$$

$$HPO_3 + H_2SO_4 \rightarrow \mathbf{2} \qquad \Delta G_r^0 = -23 kcal/mol \qquad (4)$$

The exergonic nature of reactions 3–4 suggest that formation of **2** can be spontaneous provided that suitable feedstock molecules are present. Reactions 5–7 are alternative expressions for the reactive potential of $H_3PO_4$, $SO_3$, and $HPO_3$, in which they formally combine to directly produce our model polymer **1**:

$$HPO_3 + H_2SO_4 \rightarrow \frac{1}{4}\,\mathbf{1} + H_2O \qquad \Delta G_r^0 = -21 kcal/mol \qquad (5)$$

$$HPO_3 + SO_3 \rightarrow \frac{1}{4}\,\mathbf{1} \qquad \Delta G_r^0 = -25 kcal/mol \qquad (6)$$

$$H_3PO_4 + SO_3 \rightarrow \frac{1}{4}\,\mathbf{1} + H_2O \qquad \Delta G_r^0 = -1 kcal/mol \qquad (7)$$

These estimates support the thermodynamic plausibility of forming both monomeric and polymeric phosphorus-sulfur compounds, assuming the availability of the precursor species. While the direct detection of $H_3PO_4$, $HPO_3$, or other phosphorus carriers remains elusive, earlier Vega/Venera analyses suggested trace levels of phosphorus may be present in aerosols (Andrejchikov et al., 1987), likely as phosphoric acid (Esposito et al., 2022b). More recently, the controversial detection of phosphine (Greaves et al., 2021, 2020) represent a modern line of evidence for volatile and aerosol-associated phosphorus species.

Polymerization reactions in the highly acidic Venusian cloud decks are unlikely to occur in isolation. Instead, they would be embedded within a broader network of acid-catalyzed, pH-sensitive reaction equilibria, many of which remain poorly constrained. Since developing a complete kinetic or equilibrium model of such networks lies beyond the scope of this study, we again stress the approximate nature of our predictions. Reactions 8–9 are examples of dehydration processes that could help drive polymer formation by removing water from the environment:

$$H_2O + SO_3 \rightarrow H_2SO_4 \qquad \Delta G_r^0 = -4\,kcal/mol \qquad (8)$$

$$2\,H_2O + SO_3 \rightarrow HSO_4^- + H_3O^+ \qquad \Delta G_r^0 = 4\,kcal/mol \qquad (9)$$

Notably, while reaction 9 is exergonic and proceeds without a barrier under Earth-like aqueous conditions (e.g. (Meijer and Sprik, 1998), our Venus-adjusted thermodynamic model, which incorporates low water activity corrections ($a_{water} \leq 0.0004$) (Arney et al., 2014; Donahue and Hodges, 1992; Hallsworth et al., 2021; Krasnopolsky, 2015) (see Methods), predicts it to be slightly endergonic. This difference may reflect computational limitations, such as errors in the predicted solvation energies of ions, or may be a true reflection of the extreme desiccation in the Venusian clouds. In either case, the materials described here may well be susceptible to hydrolysis. Reaction 10 represents the decomposition thermodynamics of the cyclic tetramer model **1** in the presence of water:

$$\frac{1}{4}\,\mathbf{1} + 2\,H_2O \rightarrow H_2SO_4 + H_3PO_4 \qquad \Delta G_r^0 = -3\,kcal/mol \qquad (10)$$

While reaction 10, and several other reactions calculated as spontaneous ($\Delta G < 0$), this says little about reaction rates. Indeed, thermodynamics is but one side of the coin of chemistry, the other being reactivity, kinetic stability.

### Kinetic stability and reactivity
One proxy for reactivity is a molecule's ability to accept and donate electrons. We estimate the gas phase ionization potential (IP) and electron affinity (EA) of polymer model **1** as ~11.2 eV and ~ 1.2 eV, respectively. These values indicate that polymeric phosphoric sulfonic esters are reluctant to engage in oxidative and reductive processes, reducing the likelihood of bond cleavage via homolytic or heterolytic pathways that involve radical or ionic intermediates.

To estimate the strength of the weakest S–O and P–O bond strengths in an R-substituted poly (phosphoric sulfonic ester), we turn to a simpler model, our imagined monomer, **2**:

$$\mathbf{2} \rightarrow HSO_3 + H_2PO_4 \qquad S-O\ bond \qquad \Delta H_r^0 = 60\,kcal/mol \qquad (11)$$

$$\mathbf{2} \rightarrow HSO_4^- + H_2PO_3^+ \qquad P-O\ bond \qquad \Delta H_r^0 = 51\,kcal/mol \qquad (12)$$

where reactions 11–12 correspond to homolytic and heterolytic bond cleavage, respectively. Detailed bond dissociation energies for all cleavage pathways are provided in Supplementary Table S1. Note that we use relative enthalpies as our estimate of dissociation barriers, not the relative free Gibbs energy. We do so because the entropic gain associated with an increased number of rotational and translational degrees of freedom upon dissociation is only fully realized after the bond is broken. The relative enthalpy of the separated fragments in reactions 11–12 therefore represents an upper estimate of the real Gibbs free energy barriers.

We attribute the preference for P–O cleavage, reaction 12, to a large stabilization of the formed charged fragments from the surrounding polar environment. Our estimated 50 kcal/mol dissociation energy estimate for the weakest bond of **2** is large enough to support a substantial kinetic stability of the proposed structures at temperature conditions akin to the Venusian cloud decks. We can make this inference by calculating the timescale of decomposition using the Eyring equation, assuming first-order reaction kinetics. At ambient conditions, a reaction barrier of 50 kcal/mol translates into a reaction half-life exceeding the lifetime of the universe, while at 300 °C this timescale is reduced to the order of weeks.

While the bonds inherent to these systems are predicted to be strong, polymers akin to **1** might still undergo facile decomposition through a range of more complex reaction pathways, including hydrolysis already discussed. As such, reaction pathways are likely to incorporate extensive solvent interactions, e.g., involving proton transfer and ion coordination; they are arguably best explored computationally using targeted molecular dynamics simulations – a task outside the scope of this study.

Besides solvation effects, other environmental effects could also affect reaction rates. For example, mineral aerosols or suspended particulates might catalyze bond formation just as clay minerals are thought to facilitate oligomerization of nucleotides (Ferris and Ertem, 1992) or peptide oligomerization on the early Earth (Kitadai et al., 2011). Likewise, UV-driven radical chemistry, prevalent in Venus's upper atmosphere, could promote bond formation or breakage (Mills et al., 2007). Nevertheless, as both sulfur and phosphorus are already maximally oxidized, we can assert with some confidence that oxidative degradation is unlikely.

In addition to hydrolysis and thermal cleavage, phosphate- and sulfonate-ester linkages are susceptible to transesterification, a dynamic exchange that could affect polymer persistence under Venus's cloud-deck conditions. Such exchange reactions are likely to proceed faster than complete decomposition routes, and could reshuffle chain lengths, in some cases extending and in others truncating the polymer.

Ultimately, the plausibility of polymer growth depends not only on favorable thermodynamics but also on whether reactive feedstocks attain steady-state concentrations sufficient for productive kinetics. Venusian cloud droplets have characteristic radii of ~1 μm (Hallsworth et al., 2021). For sulfur feedstocks, limitations are unlikely since the droplets themselves are concentrated sulfuric acid, and the ambient atmosphere contains $SO_2$ at ~150 ppm (Petkowski et al., 2024). In a 1 μm droplet, a single molecule corresponds to ~0.40 nM; a droplet-sized volume of gas at 150 ppm contains ~1.5 x $10^4$ $SO_2$ molecules, so incorporation of even a modest fraction yields droplet concentration in the μM range. By contrast, phosphorus availability remains uncertain; at ppb–ppm gas-phase levels, instantaneous equilibration with only a droplet-sized gas volume would imply tens of pM to tens of nM (up to low μM at the higher end). mM-level concentrations would require efficient cumulative scavenging over time, dissolution of particulate P, or in-droplet production. These estimates are necessarily speculative because steady-state values depend on poorly constrained production and loss rates, solubilities, and gas–liquid partitioning efficiencies for each feedstock species.

A summary of all studied reactions is provided in Supplementary Table S2.

### Detection characteristics
The feasibility of detecting R-substituted poly(phosphoric sulfonic esters) or related compounds in the Venusian atmosphere depends on their concentration, physical state, and spectroscopic features. The absence of conjugated π-systems in the studied molecules means that no strong absorbance is expected in the visible region.

Our electronic (UV–Vis) calculations predict peak absorbance near 160–180 nm, with very little intensity beyond 220 nm (Supplementary Figure S2). This makes the candidate polymer an unlikely contributor to Venus's mysterious UV absorber, which exhibits significant extinction in the 320–400 nm range (Titov et al., 2018). The lack of

absorption at near-visible UV wavelengths does not undermine our hypothesis that these materials may act as polypeptide analogs. In contrast, it aligns with biological backbone polymers evolving to minimize photodegradation. We note, however, that side chain substitution (i.e., identity of R) could significantly alter the photophysical properties of these molecules and polymers.

The calculated infrared (IR) spectrum of **1** (Supplementary Figure S3) features a dense array of S–O and P–O stretching modes between 400 and 1600 $cm^{-1}$, which are listed in Table 2. We focus our analysis on stretching vibrations because they offer the clearest functional-group fingerprints and occupy relatively uncluttered spectral windows in the fingerprint region. Bending modes of **1** mostly lie below 700 $cm^{-1}$, which is usually a congested window for a mixed sample, complicating comparison to reference molecules. We note that the stretching frequencies of P=O and S=O bonds of **1** that are not part of the backbone are modestly blue-shifted relative to those in phosphoric and sulfuric acid, while the P–O and S–O single bonded stretching modes belonging to the backbone are slightly red-shifted.

The polymer's most distinctive fingerprints are a pair of strong composite P–O–S stretches centered at 845 $cm^{-1}$ and 861 $cm^{-1}$; none of the reference molecules shown in Table 2 exhibits absorptions in this narrow window, making these bands useful for potentially recognizing P–O–S linkages in a mixed Venus-cloud spectrum. Note however, that while predicted relative shifts are instructive, we caution from comparing absolute stretching frequencies listed in Table 2 directly with experiment. Quantum chemical calculations naturally suffer from a range of inherent approximations, which in our case include the choice of density functional, the implicit solvent modeling, limited conformational sampling, and the harmonic approximation used to derive frequencies. Furthermore, even with high-resolution in situ instruments, overlap between multiple bands and scattering in the clouds and atmosphere is likely to hinder unambiguous spectral assignments using vibrational spectroscopy alone.

In practice, in tandem use of vibrational spectroscopy and mass spectrometry is likely the most feasible methods for detecting the envisioned materials, or related structures in the Venusian atmosphere.

**Table 2.** Calculated stretching frequencies of **1**, $H_2SO_4$, $H_3PO_4$, $SO_3$, and $SO_2$ alongside a selection of literature data

| Molecule | Assignment | Vibrational frequency ($cm^{-1}$)[a] [IR intensity][b] | | |
| | | Calc. (gas phase) | Calc. (solvated) | Literature (experimental) |
| --- | --- | --- | --- | --- |
| **1** | S–O sym. | 687 [m] | 675 [m] | |
| | S–O asym. | 716 [w] | 709 [s] | |
| | P–O asym. | 782 [w] | 768 [w] | |
| | P–O–S/P–O asym./S–O asym. | 818 [s] | 789 [s] | |
| | P–O–S/P–O sym./S–O asym. | 864 [w] | 845 [s] | |
| | P–O–S/P–O asym./S–O sym. | | 861 [s] | |
| | P–O–S/P–O sym./S–O sym. | 906 [m] | 886 [s] | |
| | S=O sym. | 1169 [s] | 1160 [s] | |
| | P=O | 1349 [s] | 1297 [s] | |
| | S=O asym. | 1414 [s] | 1380 [s] | |
| $H_2SO_4$ | S–O sym. | 767 [m] | 775 [m] | 905[c], 902[d], 834[e] |
| | S–O asym. | 825 [s] | 818 [s] | 965[c], 957[d]; 891[e] |
| | S=O sym. | 1167 [m] | 1123 [s] | 1175[c], 1158[d], 1220[e] |
| | S=O asym. | 1413 [s] | 1338 [s] | 1370[c], 1359[d], 1465[e] |
| $H_3PO_4$ | P–O sym. | 804 [w] | 821 [w] | 890[f], 885[g] |
| | P–O asym. | 895 [s] | 879 [s] | 1008[f], 1007[g] |
| | P=O | 1322 [m] | 1210 [s] | 1178f, 1165[g] |
| $SO_3$ | S=O sym. | 1019 [w] | 1018 [w] | 1065[h] |
| | S=O asym. | 1335 [m] | 1313 [s] | 1391[h] |
| $SO_2$ | S=O sym. | 1132 [w] | 1134 [w] | 1147[i], 1150[h] |
| | S=O asym. | 1314 [m] | 1287 [s] | 1351[i], 1350[h] |

[a]B3LYP-D3(BJ)/6–31 + G(d,p) (unscaled).
[b]IR intensity shown as w = weak (<100 a.u.), m = medium (100–300 a.u.), and s = strong (>300 a.u.).
[c]Transmission IR of aqueous $H_2SO_4$ at room temperature (Walrafen and Dodd, 1961).
[d]ATR-IR spectrum of 18 M liquid $H_2SO_4$ at room temperature (Horn and Jessica Sully, 1999).
[e]Vapor phase IR spectra of $H_2SO_4$ at 150 ℃ (Hintze et al., 2003).
[f]Aqueous $H_3PO_4$ at 23 ℃ (Rudolph, 2010).
[g]Aqueous $H_3PO_4$ (Chapman and Thirlwell, 1964).
[h]NIST.
[i]Experiment (Eigner et al., 2009).

## Relevance to Venus missions

Can polymers akin to **1** be detected by planned missions to Venus? Unlikely. Detecting non-volatile P–O–S polymers like **1** will be exceptionally challenging for the next generation of Venus missions. As summarized in Supplementary Table S3, most payloads are optimized for volatile gases and surface mineralogy; none include mid-IR or Raman capabilities needed to resolve the diagnostic P=O, S=O, or P–O–S stretches of condensed-phase materials. Quadrupole mass spectrometers slated to fly (e.g., on DAVINCI) could register low-mass fragments or monomers, but unit-mass resolution and reliance on hard electron-ionization severely limit recognition of intact macromolecules. In short, present instruments favor gas-phase species and leave a sensitivity gap for complex polymers, highlighting the need for future soft-ionization mass spectrometry or mid-IR spectrometers if such materials are to be detected directly. Including a mid-IR spectrometer could prove particularly worthwhile, given the relatively small mass cost.

## Conclusion

This work constitutes a computational exploration of hypothetical sulfur- and phosphorus-rich polymers as polypeptide analogs in Venus's acidic cloud decks. The aim is not to propose details for the emergence of alternative heteropolymers on Venus, nor to argue that P–O–S-based polymers are the most likely materials of biological relevance in such environments. Rather, we share an example of computational astrobiology meant to illustrate how quantum chemical modeling can be used to evaluate the basic feasibility of hypothetical biopolymer analogs in exotic environments.

Using quantum chemical methods, we have shown that R-substituted poly(phosphoric sulfonic esters) could form spontaneously from plausible atmospheric constituents under Venusian conditions, and that such materials may be sufficiently kinetically stable to persist in the lower cloud decks. Their formation compares favorably to polypeptides in polar solution, and their predicted vibrational spectra exhibit characteristic P–O–S stretching modes that could serve as a potential diagnostic marker. We unfortunately conclude that currently planned missions to Venus are unlikely to conclusively detect these kinds of materials, would they be present, or, indeed, any macromolecular species of potential biological relevance. Future instrumentation capable of mid-IR or Raman spectroscopy, or soft ionization mass spectrometry, would likely be necessary for probing the molecular signatures of non-volatile sulfur/phosphorus-based polymers.

The use of quantum chemical calculations to vet speculative ideas before committing to resource-intensive modeling or experiments is attractive. However, deciding which hypotheses to pursue and to what depth is not trivial. While our study outlines basic stability and detection characteristics, a more comprehensive assessment of our target polymer's persistence (e.g., resistance to hydrolysis) would require kinetic modeling of formation and degradation pathways alongside dynamic acid–base equilibria under ultralow water activity. Additional variables such as cation coordination, backbone or side-chain driven folding, protonation-state dependent reactivity, and the prospect of enzyme-like catalysis in concentrated sulfuric acid further complicate the picture, far beyond the scope of this study.

More broadly, this work reiterates the value of reframing our search for life by challenging carbon chauvinism – the assumption that life must resemble Earth-based carbon biochemistry. By illustrating how plausible, metastable biopolymer analogs could emerge and persist in a radically different environment, we emphasize the need to expand our chemical imagination in the search for agnostic biosignatures. Venus, like Saturn's moon Titan, can serve as a test bed for exploring the limits of alternative chemistries (e.g., (Rahm et al., 2016; H. Sandström and Rahm, 2020; Stevenson et al., 2015), reminding us that the search for life should look beyond familiar paradigms. Embracing this broader lens means prioritizing more agnostic astrobiology mission payloads – tools capable of detecting the unexpected – so that when we peer into alien environments, we are equipped to recognize life, however strange its chemistry may be.

## Methods

All calculations were performed using the hybrid-exchange-correlation functional B3LYP (Becke, 3-parameter, Lee–Yang–Parr) (Becke, 1993; Lee et al., 1988), combined with the D3BJ dispersion correction (Grimme et al., 2010, 2011) and the 6–31 + G(d,p) basis set, as implemented in Gaussian 16, revision B.01 (Frisch et al., 2017). Solvent effects were described implicitly using the Solvation Model based on Density (SMD) (Marenich et al., 2009) parametrized for water, but with the dielectric constant modified to 100, approximating that of concentrated sulfuric acid (Gillespie and Cole, 1956). Our implicit modeling of solvation effects neglects explicit entropic and enthalpic contributions from solvent structuring and ion coordination, and it cannot capture local solvent reorganization around reactive groups. More advanced treatments, such as QM/MM molecular dynamics simulations, could account for these effects more rigorously, but remain outside the scope of the present exploratory study.

Conformational sampling was performed using AutodE (Young et al., 2021). Frequency analyses were performed on all structures to verify the absence of imaginary modes, to evaluate thermal and entropic corrections to the electronic energy, and to generate infrared (IR) spectra. Ultraviolet–visible (UV–Vis) spectra were calculated using time-dependent (TD) DFT (Runge and Gross, 1984), with excitation energies obtained from the lowest 50 singlet and triplet states.

Gibbs free energies are reported for 298.15 K and corrected to standard state concentrations of 1 M for all species, except for sulfuric acid and water, which are referenced to conditions representative of the Venusian cloud decks. We adopt a sulfuric acid concentration of 16.3 M and a water activity of 0.000533 (dimensionless), based on the cloud droplet composition at 25 °C reported by Hallsworth et al. (2021). While atmospheric conditions on Venus vary with altitude, this midpoint estimate provides a chemically relevant reference point for assessing thermodynamic feasibility.

Gibbs energies corrections for non-gas-phase standard states were obtained using the expression:

$$\Delta G_{\text{corrected}} = \Delta G^0(1\,atm) + RT\sum_j v_j \ln a_j$$

$$= \Delta G^0(1\,atm) + RT\sum_j v_j \ln \frac{c_j^{\text{target}}}{c^{\text{gas}}}$$

where $\Delta G^o(1\,atm)$ is the sum of electronic and thermal free energies obtained from a frequency analysis, $R$ is the gas constant ($1.987 \times 10^{-3}$ kcal mol$^{-1}$ K$^{-1}$), $T$ is the temperature (298.15 K), $v_j$ is the stoichiometric coefficient of species $j$ (positive for products,

negative for reactants), $a_j$ is the activity of species $j$, $c_j^{\text{target}}$ is the chosen standard concentration for species $j$, and $c^{\text{gas}} = 1$ atm/ RT = 0.0409 M at 298.15 K.

**Open peer review.** To view the open peer review materials for this article, please visit http://doi.org/10.1017/qrd.2025.10012.

**Supplementary material.** The supplementary material for this article can be found at http://doi.org/10.1017/qrd.2025.10012.

**Data availability statement.** All raw computational output files with necessary numerical data to reproduce any of the calculations in this work are openly available on Zenodo at https://doi.org/10.5281/zenodo.15881845.

**Acknowledgements.** We acknowledge the Purdue Community Cluster Program as described in (McCartney et al., 2014). We gratefully acknowledge the constructive feedback provided by the four anonymous reviewers, which strengthened the quality of this work.

**Author contribution.** Conceptualization: MR, Writing – review and editing: IM, MR, SAA, JH, Supervision of research: MR, FIR, HS. Investigation & Formal Analysis: IM, SAA, JH, EM, EH, KK, FN, AS, FIR, HS, MR.

**Financial support.** We thank Chalmers Gender Initiative for Excellence, Chalmers Area of Advance Nano, and Jonathan Tan and the CASSUM fellowship, for funding parts of this research. Our research relied on computational resources provided by the National Academic Infrastructure for Supercomputing in Sweden (NAISS) at C3SE and NSC partially funded by the Swedish research council through grant agreement no. 2022–06725 and in part through the Negishi cluster, operated by the Rosen Center for Advanced Computing at Purdue University.

**Competing interest.** The authors declare none.

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
