## [Reviewer Report]

Review of “On the Possibility of Carbon-Free Biopolymers on Venus: A Computational Astrobiology Study”

It is speculated that the atmosphere of Venus might contain polymeric sulfur and phosphorus-based oxides which act as biopolymers. This speculation about the existence of such compounds is carefully supported by computations pointing to the existence of such compounds. I favor publication of this study, but I am not at ease with the choice of title. The authors favor the title “On the Possibility of Carbon-Free Biopolymers on Venus: A Computational Astrobiology Study.” I think what this study shows is the possibility of polypeptide analogs, but it is not clear what, if any, biological activity they might have. Therefore, I am reluctant that these compounds be referred to as biopolymers. I suggest that biopolymers not be in the title of the paper by itself but rather something like possible analogs of biopolymers.

---

## [Reviewer Report]

The paper presents an analysis and discussion of the possibility of development of life on Venous. The issues explored are very intriguing and interesting and should be of interest to the readers . However, the background and the methods used for the actual calculations may not be up to the current state of the art treatments.

My concerns include the argument that the current estimates do not reflect free energies since the chamges in entropic contributions of bond rotations are not expected to be large. However, the main issue is the entropic contributions of the sourounding solvent, In fact, proper calculations should have involved QM/MM free energy calculations that include the assumed polar environments ( e.g. J. Phys. Chem. B 2013, 117 (42), 12807–12819. https://doi.org/10.1021/jp4020146

I realize that such calculations are challenging but the readers must be exposed to the possible problems with the current estimates.

---

## [Reviewer Report]

Madan et al. carried out quantum chemical calculations to assess thermodynamic parameters for the formation and decomposition of cyclic heteropolymers consisting of phosphoric-sulfonic diester subunits under conditions similar to those of the atmospheric cloud decks of Venus. Venusian clouds at high altitude consist of sulfuric acids aerosols with low water content at moderate temperature and pressure. The authors’ modeling suggests that such polymers could form and persist, and thus potentially provide the basis for informational heteropolymers that are thought to be essential for life. They also suggest what distinct wavelengths might be observed in the “signature region” of the infrared spectrum to indicate that such compounds indeed exist in Venusian clouds, should a suitable detection system become available. This is a jaunty manuscript that would be suitable for publication in QRB Discovery, subject to the minor revisions described below.

1. It is not appropriate to refer to the polyphosphoric-sulfonic compounds as “biopolymers”. The central point of this study is that these polymers could plausibly form spontaneously under abiotic conditions similar to those of the atmospheric cloud decks of Venus. Thus, these polymers cannot be regarded as potential biosignatures, but rather agnostic signatures that may reflect either abiotic or biotic processes (for extensive discussion, see An Astrobiology Strategy for the Search for Life in the Universe, National Academies Press, Washington DC, 2019, Ch. 4). A better term would be “heteropolymer”.

2. Although there are some superficial structural similarities between the polyphosphoric-sulfonic compounds and polypeptides, as shown in Figure 1, there are important differences that likely will make the former much less amenable to exhibiting function. There are eight rotatable bonds within each phosphoric-sulfonic subunit of the hypothetical heteropolymer compared to only three rotatable bonds within each amino acid subunit of a polypeptide. The side chains of the hypothetical heteropolymer are more likely to disrupt the backbone structure and perturb polymerization efficiency compared to the side chains of polypeptides. A comparison to teichoic acids might be more apt. In any case, the authors need to tone down the comparative reference to polypeptides.

3. The authors’ discussion of the “phosphate problem” neglects to reference the recent work of Catling and colleagues showing that phosphate can accumulate to high levels within carbonate-rich lakes (see Toner & Catling, Proc Natl Acad Sci USA, 117, 883–888, 2020).

4. The term “firebrand” (line 178) has the connotation of “agitator”, which is not likely what the authors intended. A better word would be “advocate”.

5. The decision to model cyclic as opposed to linear heteropolymers is understandable as a practical matter, but the authors need to acknowledge the effect this decision may have on their results. Dimer formation may be hindered due to the need to close the macrocycle, whereas subsequent addition reactions may benefit from relieving ring strain. This is another way in which the thermodynamic profile does not “resemble that of terrestrial peptide synthesis” (line 261), which is a linear process. There is an extensive literature on cyclic peptide synthesis, both non-enzymatic and through non-ribosomal peptide synthesis, with the former making use of activating groups and the latter catalyzed by biological enzymes.

6. The authors need to discuss plausible steady-state concentrations for the monomeric building blocks in Venusian clouds. Of course this would be speculative, but it would be beneficial to consider what concentrations would be needed to achieve productive kinetic rates for polymer synthesis.

7. The authors need to consider phosphoester and sulfoester transesterification that would result in disproportionation reactions that both increase and decrease polymer chain length. These reactions likely would occur at a much faster rate than polymer decomposition.

8. It is difficult to understand why such a simple study requires 11 co-authors. The senior author should provide a statement regarding the contributions of each author and consider whether some of the current authors should instead be acknowledged rather than given co-authorship.

---

## [Reviewer Report]

I really enjoyed reading the paper. the paper might be a step forward to answer a fascinating question: can life exist without water?

The authors explore the possibility of life in acidic cloud decks of Venus’ atmosphere. They explore carbon-free biopolymer that might be sourced from feedstock in the Venusian atmosphere.

The manuscript is well written and should be disseminated in the scientific literature.

I have a curiosity. To my knowledge Venus atmosphere is mainly composed of carbon dioxide (approximately 96.5%) and nitrogen (around 3.5%) plus some ppm of other gases, including sulfur based compounds,160 ppm. Can the chemical pathways described in the paper be active at so low sulfur derivatives concentrations?